# Reverse-engineering Recurrent Neural Network solutions to a hierarchical inference task for mice

**Rylan Schaeffer**
Institute for Applied Computational Science
Harvard University
rylanschaeffer@g.harvard.edu

**Mikail Khona**
Department of Physics
Massachusetts Institute of Technology
mikail@mit.edu

**Leenoy Meshulam**
Department of Brain and Cognitive Sciences
Massachusetts Institute of Technology
leenoy@mit.edu

**International Brain Laboratory**
info@internationalbrainlab.org

**Ila Rani Fiete**
Department of Brain and Cognitive Sciences
McGovern Institute for Brain Research
Massachusetts Institute of Technology
fiete@mit.edu

## Abstract

We study how recurrent neural networks (RNNs) solve a hierarchical inference task involving two latent variables and disparate timescales separated by 1-2 orders of magnitude. The task is of interest to the International Brain Laboratory, a global collaboration of experimental and theoretical neuroscientists studying how the mammalian brain generates behavior. We make four discoveries. First, RNNs learn behavior that is quantitatively similar to ideal Bayesian baselines. Second, RNNs perform inference by learning a two-dimensional subspace defining beliefs about the latent variables. Third, the geometry of RNN dynamics reflects an induced coupling between the two separate inference processes necessary to solve the task. Fourth, we perform model compression through a novel form of knowledge distillation on hidden representations – Representations and Dynamics Distillation (RADD) – to reduce the RNN dynamics to a low-dimensional, highly interpretable model. This technique promises a useful tool for interpretability of high dimensional nonlinear dynamical systems. Altogether, this work yields predictions to guide exploration and analysis of mouse neural data and circuity.

## 1 Introduction

Making decisions involves weighing mutually-exclusive options and choosing the best among them. Selecting the optimal action requires integrating data over time and combining it with prior information in a Bayesian sense. Here we seek to understand how RNNs perform hierarchical inference. For concreteness and for the later goal of comparing against the mammalian brain, we consider a perceptual decision-making and change-point detection task used by the International Brain Laboratory (IBL) [1], a collaboration of twenty-two experimental and theoretical neuroscience laboratories. Optimally solving the IBL task requires using sensory data to infer two latent variables, one cued and one uncued, over two timescales separated by 1-2 orders of magnitude.

We address two questions. First, how do RNNs compare against normative Bayesian baselines on this task, and second, what are the representations, dynamics and mechanisms RNNs employ to perform inference in this task? These questions are of interest to both the neuroscience and the machine learning communities. To neuroscience, RNNs are neurally-plausible mechanistic models that can serve as a good comparison with animal behavior and neural data, as well as a source of scientific hypotheses [17, 42, 5, 36, 10, 8]. To machine learning, we build on prior work reverse engineering how RNNs solve tasks [41, 38, 18, 17, 3, 25, 16, 24], by studying a complicated task that nevertheless has exact Bayesian baselines for comparison, and by contributing task-agnostic analysis techniques.

The IBL task is described in prior work [40], so we include only a brief summary here. On each *trial*, the mouse is shown a (low or medium contrast) stimulus in its left or right visual fields and must indicate on which side it perceived the stimulus. Upon choosing the correct side, it receives a small reward. Over a number of consecutive trials (a *block*), the stimulus has a higher probability of appearing on one side (left stimulus probability $p_s$, right stimulus probability $1 - p_s$). In the next block, the stimulus side probabilities switch. The change-points between blocks are not signaled to the mouse. This experimental block design is occasionally called change-point detection [2, 27].

## 2 Methods

### 2.1 IBL task implementation

Each session consists of a variable number of trials, indexed $n$. Each trial is part of a block, with blocks defining the prior probability that a stimulus presented on the trial is shown on the left versus the right. The block side on trial $n$, denoted $b_n \in \{-1, 1\}$ (-1: left, 1: right), is determined by a 2-state semi-Markov chain with a symmetric transition matrix. The probability of switching block sides between two consecutive trials is $p_b$ and the probability of remaining is $1 - p_b$. The process is semi-Markov because $p_b$ varies as a function of the current block length ($l_n$) to ensures a minimum block length of 20 and maximum of 100, with otherwise geometrically distributed block lengths.

$$\begin{bmatrix} P(b_n = 1) \\ P(b_n = -1) \end{bmatrix} = \begin{bmatrix} 1 - p_b & p_b \\ p_b & 1 - p_b \end{bmatrix} \begin{bmatrix} P(b_{n-1} = 1) \\ P(b_{n-1} = -1) \end{bmatrix} \qquad p_b = \begin{cases} 0 & l_n < 20 \\ p_{b0} & 20 \leq l_n \leq 100 \\ 1 & 100 < l_n \end{cases}$$

The stimulus ($s_n \in \{-1, 1\}$) presented on trial $n$ is either a left or right stimulus, determined by a Bernoulli process with a single fixed parameter $p_s$, which gives the probability that the stimulus is on the same side as the current block (termed a *concordant* trial). The probability of a *discordant* trial (stimulus on opposite side of block) is $1 - p_s$. In the IBL task, $p_{b0} = 0.02$ and $p_s = 0.8$.

Neural time-constants are much shorter than the timescale of trials, so we model a trial as itself consisting of multiple timesteps indexed by $t$. A trial terminates if the RNN takes an action (explained in the next subsection) or if the RNN fails to take an action after 10 steps. At the start of the trial, the stimulus side $s_n$ and a stimulus contrast $\mu_n$ are sampled (Fig. 1). Within a trial, on each step, the RNN receives three scalar inputs. On the first step, all three are 0. For each subsequent step, the RNN receives two noisy observations $o_{n,t}^L, o_{n,t}^R$, sampled i.i.d. from two univariate Gaussians with mean

$$b_n \sim P(b | b_{n-1}, p_b(l_n))$$
$$s_n = b_n | b_n \sim Bern(p_s)$$
$$\mu_n \sim \mathcal{U}([0, 0.5, 1.0, 1.5, 2.0, 2.5])$$
$$o_{n,t}^S | \mu_n \sim \mathcal{N}(\mu_n, 1)$$
$$o_{n,t}^{\sim S} | \mu_n \sim \mathcal{N}(0, 1)$$

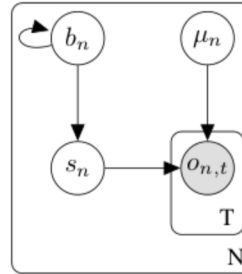

Figure 1: Generative model of the IBL task. Block side $b_n$ is determined by a 2-state semi-Markov chain. Stimulus side $s_n$ is either $b_n$ with probability $p_s$ or $-b_n$ with probability $1 - p_s$. Trial stimulus contrast $\mu_n$ determines the observations on each timestep $t$ within a trial: $o_{n,t}^S$ for the stimulus side, $o_{n,t}^{\sim S}$ for the non-stimulus side.

$\mu_n$ for the stimulus side and mean 0 for the other. The third input is a feedback signal $r_{n,t}$, which takes one of three possible values: a small waiting penalty (-0.05) in every timestep, a reward (+1) if the correct action was taken on the previous step, or a punishment (-1) if the incorrect action was taken on the previous step or the model failed to act.

## 2.2 Recurrent network architecture and training

On each step, the observation $o_{n,t} = \begin{bmatrix} o_{n,t}^L & o_{n,t}^R & r_{n,t} \end{bmatrix}^T$ is input to the RNN. The RNN state $h_{n,t}$ on the $n$th trial and the $t$th step within the trial is defined by the typical dynamics:

$$h_{n,t} = \tanh(W^{rec}h_{n,t-1} + W^{in}o_{n,t} + b^{rec})$$
$$a_{n,t} = \text{softmax}(W^{out}h_{n,t} + b^{out})$$

where $a_{n,t}$ is a probability distribution over the two possible actions (left or right) and the state $h_{n,t}$ at the end of one trial is the state at the start of the next trial. An action is defined by when the probability mass on either action exceeds a fixed threshold (0.9). We present RNNs with 50 hidden units, but the results are similar for other numbers of units (SI 7.5). The RNNs are trained to minimize cross entropy integrated across all RNN steps with the stimulus side as the target distribution. For a more detailed description of the training setup, see SI 7.2. Code is publicly available at `https://github.com/int-brain-lab/ann-rnns`.

## 2.3 Normative Bayesian baselines

The IBL task involves inference of two latent variables, the stimulus side and the block side. Exact inference can be decomposed into two inference subproblems that occur over different timescales, which we term *stimulus side inference* and *block side inference*:

$$\underbrace{P(s_n|s_{<n}, o_{\leq n, \leq T})}_{\text{Current stimulus posterior}} = \underbrace{\frac{P(o_{n,\leq T}|s_n)}{P(o_{n,\leq T})}}_{\text{Stimulus side inference}} \underbrace{P(s_n|s_{\leq n-1}, o_{\leq n-1, \leq T})}_{\text{Block side inference}}$$

where $\cdot_{\leq m}$ denotes all indices from 1 to $m$, inclusive. *Stimulus side inference* occurs at the timescale of a single trial. Since observations within a trial are sampled i.i.d., the observations are conditionally independent given the trial stimulus contrast $\mu_n$. The likelihood is therefore:

$$P(o_{n,\leq T}|s_n) = \sum_{\mu_n} P(o_{n,\leq T}|\mu_n)P(\mu_n|s_n) = \sum_{\mu_n}\left(\prod_{t=1}^{T} P(o_{n,t}|\mu_n)\right)P(\mu_n|s_n)$$

*Block side inference* occurs at the timescale of blocks, based on knowledge of the history of true stimuli sides. Our Bayesian baselines assume that the block transitions are Markov (instead of semi-Markov). Both baselines perform Bayesian filtering [32] to compute the block side posterior by alternating between a joint and a conditional, and normalizing after each trial:

$$P(b_n, s_n|s_{\leq n-1}) = \sum_{b_{n-1}} P(s_n|b_n)P(b_n|b_{n-1})P(b_{n-1}|s_{\leq n-1})$$
$$P(b_n|s_{\leq n}) = \frac{P(b_n, s_n|s_{\leq n-1})}{\sum_{b_n} P(b_n, s_n|s_{\leq n-1})}$$

We consider two Bayesian baselines. The *Bayesian actor* performs the task independently from the RNN but uses the same action rule (i.e. an action taken when its stimulus posterior passes the action threshold 0.9). The *Bayesian observer* receives the same observations as the RNN, but does not choose when to act: the RNN determines how long a given trial lasts. The Bayesian actor tells us what ceiling performance is, while the Bayesian observer upper-bounds how well the RNN could have done given the inputs, at the time the RNN chooses to act.

The Bayesian actor and the Bayesian observer are otherwise identical. Both assume perfect knowledge of the task structure and task parameters, and both are comprised of two separate submodels performing inference. The first submodel performs stimulus side inference given the block side, while the other submodel infers block changepoints given the history of true stimuli sides. The true stimulus side can be determined after receiving feedback about whether the selected action was correct (so long as an action was selected).

# 3 Results

## 3.1 RNN Behavior

### 3.1.1 RNN behavior matches ideal Bayesian observer behavior

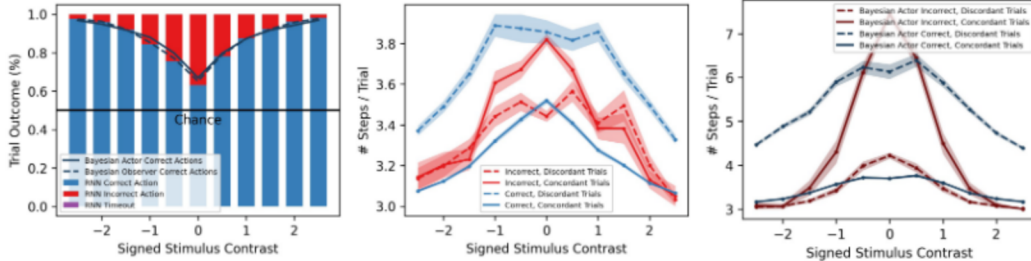

Figure 2: (a) RNN performance (fraction of actions that are correct) almost matches Bayesian observer and Bayesian actor. (b) RNN chronometric curves show longer integration on low contrast trials. (c) Bayesian actor chronometric curves show the Bayesian actor integrates for significantly longer on low contrast trials than the RNN.

We start by quantifying the performance of the RNN. Strikingly, the RNN achieves performance nearly matching the Bayesian observer (Fig. 2a). Both display similar accuracy to the Bayesian actor as a function of stimulus contrast $\mu_n$: the fraction of correct actions is highest for high stimulus contrast and lowest for zero contrast. Furthermore, the performance of all three agents is well above chance (50%) for zero-contrast trials, showing that all three exploit prior evidence from the block structure of the task (we randomly assign a correct stimulus side for each trial based on the block probability, even when stimulus contrast is zero).

Chronometric curves, which quantify how quickly the agents select an action as a function of stimulus contrast, show that both the RNN and the Bayesian actor respond faster on *concordant* trials (when the stimulus side matches the block side, an event with higher prior probability) than *discordant* trials (Figs. 2bc). While both the RNN and the Bayesian agents act more slowly (accumulating evidence for longer) on low stimulus contrast trials, the RNN acts significantly faster than the Bayesian actor on trials with low stimulus contrast (Figs. 2bc). Despite a significantly shorter integration time on the lowest-contrast stimuli, RNN behavior is only slightly suboptimal (with respect to trial accuracy) than the Bayesian actor, suggesting that longer integration of inputs on near-zero stimulus contrast trials offers marginal benefits. This suggests that a reward maximizing agent might well choose a strategy more similar to the RNN than to an accuracy maximizing agent like the Bayesian actor.

### 3.1.2 RNN leverages block prior when selecting actions

We next explored to what extent the RNN leverages the block prior to select actions. The RNN and the Bayesian actor both perform much better on concordant trials than discordant trials (Fig. 3a), direct evidence of the influence of estimated block side on trial-by-trial decision making. The concordant-discordant discrepancy shrinks with increasing stimulus contrast, meaning that when the contrast is high, the stimulus (likelihood) dominates over the block prior,while at low contrasts, the block prior dominates. The closeness of RNN performance to that of the ideal Bayesian actor suggests that the RNN estimates and combines stimulus side likelihood and block prior near-optimally.

However, there are two small differences that suggest the RNN slightly underweights the prior. First, the concordant-discordant gap is smaller in the RNN than the Bayesian actor. Second, on zero-contrast trials, the Bayesian actor's accuracy directly reflects the block prior (.2/.8), while the RNN's accuracy is slightly contracted towards chance performance (.5/.5). This is likely not due to deficiencies in the RNN's inference of the block side, as the RNN's fraction of correct answers rapidly climbs following a block change-point in a way that closely matches the Bayesian baselines (Fig. 3b). We confirm the RNN tracks the block prior by noting that, as expected after a block changepoint, the fraction of correct trials for zero contrast trials climbs from above 0.2 to below 0.8 (Fig. 3c).

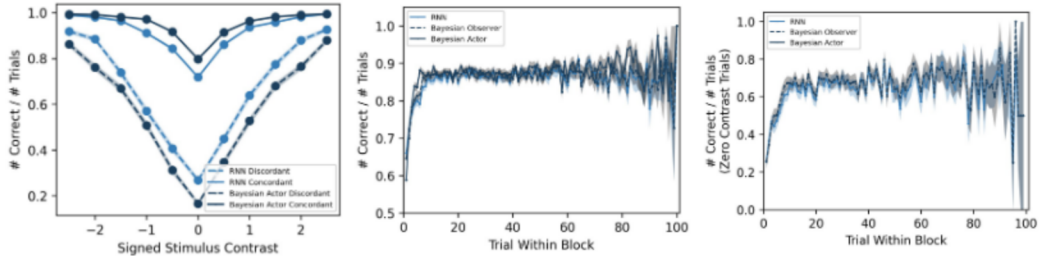

Figure 3: (a) Performance psychometric curves for RNN and Bayesian actor split by whether the current stimulus is on same (concordant) or oppposite (discordant) side relative to the current block. The split shows the influence of the inferred block prior; the Bayesian actor-RNN match shows the RNN near-optimally incorporates the block prior. (b) RNN fraction of correct trials following a block change-point rapidly increases, closely matching the Bayesian observer. (c) Fraction of correct trials for zero contrast trials rises from nearly 0.2 to 0.8 (consistent with the block priors).

## 3.2 RNN Representations

### 3.2.1 RNN learns 2D dynamics to encode stimulus side and block side

We next sought to characterize how the RNN's representations and dynamics subserve inference. The first two principal components (PCs) of RNN activity explain 88.74% of the variance in activity, suggesting the RNN has learned a low-dimensional dynamical solution. The RNN readout vector, which converts hidden states into actions, explicitly gives us the direction along which the RNN encodes its stimulus side belief; we term this the *stimulus readout* vector. 93.39% of the stimulus readout vector's length lies within the 2D PCA plane.

To identify how block side is encoded in RNN activity, we trained a logistic classifier to predict block side at each RNN step. This classifier had 82.6% accuracy on a 67-33% train-test split. A separate classifier for block side trained from the 2-dimensional PCA plane of RNN activity had 82.5% accuracy (Fig. 4a), affirming that the RNN's PCA plane encompasses the two latent variables being inferred. These two dimensions are sufficient to decipher how the network solves the task. Importantly, the block and stimulus readouts are non-orthogonal (subtending an angle of $73°$ in the high-dimensional RNN space, and $68°$ in the PCA plane). This deviation from orthogonality is modest but critical to how the network performs hierarchical inference (as we explain below).

State-space trajectories (Fig. 4bc) in the PCA plane reveal how the RNN state evolves across trials in a block. Each point is the RNN state at the end of a trial. Right after a block changepoint, the state begins on the wrong side of the block decision boundary. Trial-by-trial, the state appropriately

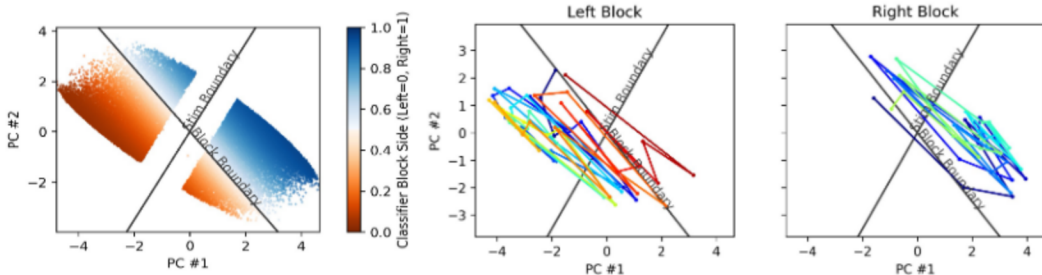

Figure 4: (a) Logistic regression predicts block side from RNN activity with 82.6% accuracy on a 67-33% train-test split. Each point is RNN activity at a different step within a session. Non-orthogonal decision boundaries show clear separation along the stimulus readout but a murkier distinction along the block readout. (b-c) The evolution of RNN states across trials in left (b) and right (c) blocks. Color: trial number within the block (blue=early, red=late). RNN states jump back and forth across the stimulus decision boundary while slowly progressing toward the correct block direction.

crosses the stimulus boundary to encode the stimulus side. Over multiple trials, it gradually advances along the block readout vector to transition from the previous block side to the new block side.

### 3.2.2 Observations are integrated to infer stimulus side and block side

Based on state-space trajectories, we hypothesized that the RNN infers the stimulus side and block side by integrating observations at different rates, faster (slower) timescales for stimulus (block) inference. We confirmed this by plotting the velocity of the RNN state along the stimulus and block readout vectors, as a function of the difference in the right and left observation values $d_{n,t} = o_{n,t}^R - o_{n,t}^L$ (Fig. 5ab). Both had positive slopes (0.84 for stimulus, 0.18 for block) with $p < 1e - 5$, confirming that rightward (leftward) stimuli simultaneously move the state both toward the right (left) side of the stimulus decision boundary and toward the right (left) side of the block decision boundary. The respective magnitude of these two slopes (stimulus slope $\approx 5 *$ block slope) match our expectation that stimulus side inference changes more rapidly with observations than block side inference.

The RNN integrates these instantaneous state velocities to perform inference of stimulus and block side. The component of RNN activity along the right block readout vector closely matches the average block side posterior estimate of the Bayesian observer/actor (Fig. 5c, curves averaged across trials and blocks), up to an arbitrary scaling parameter (determined through an ordinary least squares fit between the two; the Bayesian actor and observer are identical for block side inference). This result reveals how the RNN performs efficient change-point detection of block side.

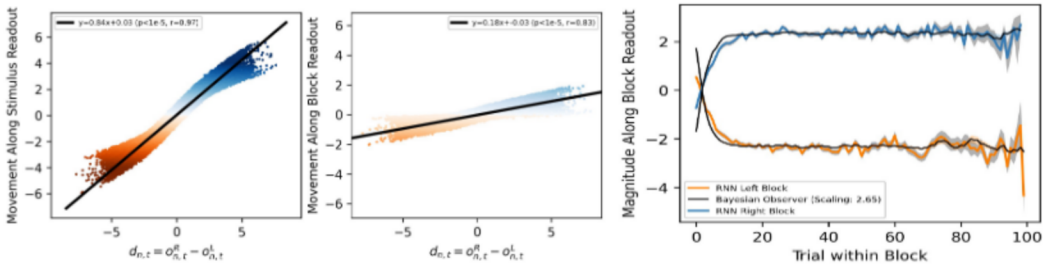

Figure 5: (a-b) Left (right) sensory evidence pushes the RNN state toward the left (right) stimulus and left (right) block readouts, but with amplitudes that differ by an order of magnitude (slopes). (c) The instantaneous changes along block readout direction (b) are integrated to infer block side.

On a trial-by-trial basis, however, the RNN block belief has larger-amplitude fluctuations (Fig. 6a) than the Bayesian observer's block posterior. These fluctuations are driven directly by trial-by-trial stimuli: the stimulus side on each trial has a larger effect on the RNN's block belief than on the Bayesian observer's block posterior (Fig. 6b). This discrepancy is due to an induced coupling in the RNN between stimulus and block inference: the RNN updates its block and stimulus beliefs

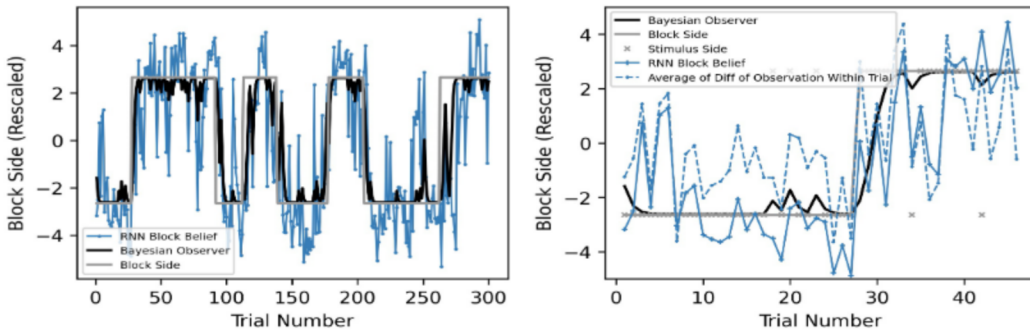

Figure 6: (a) The RNN's belief about block side closely tracks both the Bayesian observer's block posterior and the true block side, albeit with larger short-term fluctuations. (b) Fluctuations in the RNN's block belief are due to trials with large jumps in evidence, given by $o_{n,t}^R - o_{n,t}^L$ (dashed line).

simultaneously at each step and cannot decouple the two inference problems, whereas the Bayesian approach decouples the two inference problems and controls *when* information is communicated.

### 3.3 RNN Mechanism

#### 3.3.1 RNN dynamics and connectivity are consistent with bistable/line-attractor dynamics

Visualizing flows of the RNN in the state space (Fig. 7) reveals the behavior of the system. In the presence of a stimulus, the network states flow toward one of two discrete attractors, in the right-block right-stimulus quadrant or in the left-block left-stimulus quadrant. When the stimulus is absent and there is no performance feedback, the network exhibits a 1-dimensional line attractor. The line attractor is mainly aligned with the block readout vector, which allows the RNN to preserve its block side belief across trials. Even though block side is itself a discrete variable (left or right), the block posterior and the network's estimate of block side are, and should be, continuous quantities. The line attractor has a small projection along the stimulus readout vector, which translates the block belief into a stimulus prior, in the form of an initial condition for the state on the next trial that biases the

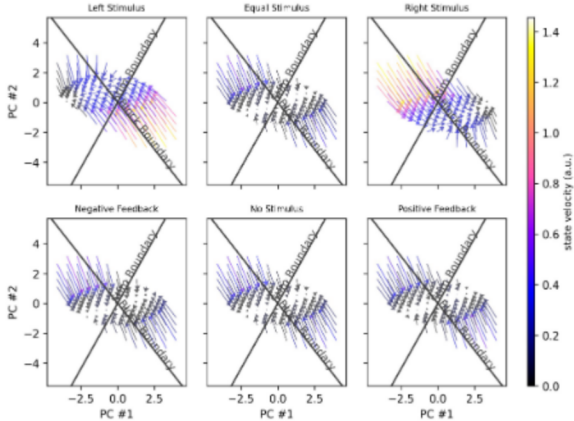

Figure 7: Flows (one-step changes) of the RNN states plotted in state space, under six input conditions. Lines indicate direction of flow; color indicates magnitude (speed) (black = low, yellow = high).

RNN to select the concordant stimulus side in its decision. Surprisingly, performance feedback has little effect on network state (Fig. 7), possibly because combining feedback with the chosen action to determine the correct action is non-trivial to learn, or because the trained RNN's actions are typically correct so performance feedback conveys little additional information.

What are the circuit mechanisms responsible for the RNN's cpmputation? The state-space portraits of integration to two discrete fixed points (trial decision) and integration along a line attractor (block inference) are reminiscent of a particular circuit motif in the brain capable of producing 1-dimensional line-attractor dynamics or bistable attractor dynamics, depending on the strength of the excitatory and inhibitory recurrent connections [33, 23, 44, 20, 28]. This circuit motif has been studied in tasks involving a single variable, but not in tasks involving two (hierarchically interacting) variables.

#### 3.3.2 Model extraction by distillation of hidden unit representations and dynamics

To extract a low-dimensional, interpretable model of the RNN and reveal its effective circuit, we propose a variation of knowledge distillation [7, 4, 14] in which we train a small RNN with hidden states $\hat{z}_t$ to reproduce a low-dimensional projection of the *hidden states* of the original RNN. This differs from conventional distillation in which the small network is trained on the *output* probabilities or logits of the original model (although a similar technique was used in BERT Transformer networks for NLP [15, 13]). We call our approach *Representation and Dynamics Distillation* (RADD). Specifically, we train the parameters $A'$, $B'$, $c'$ of a smaller RNN so its states $\hat{z}_t$ recapitulate a projection of the original RNN's hidden states ($\{z_t \equiv Ph_t\}_{t=1}^T$, starting from initial condition $\hat{z}_1 = z_1 = Ph_1$, where $P$ is the $M \times N$-dimensional dimension-reducing projection matrix. After selecting $P$, the small RNN is trained simply via regression (additional details in SI 7.3):

$$\operatorname*{arg\,min}_{A',B',c'} \sum_{t=1}^T ||z_t - f(A'\hat{z}_{t-1} + B'o_t + c')||^2.$$

#### 3.3.3 Reduced model preserves RNN dynamics and recovers meaningful parameters

Because the RNN dynamics are well-captured by two principal components, we hypothesized that it might be possible to reduce it to a mere 2-unit RNN. Indeed, the dynamics of a 2-unit RADD RNN

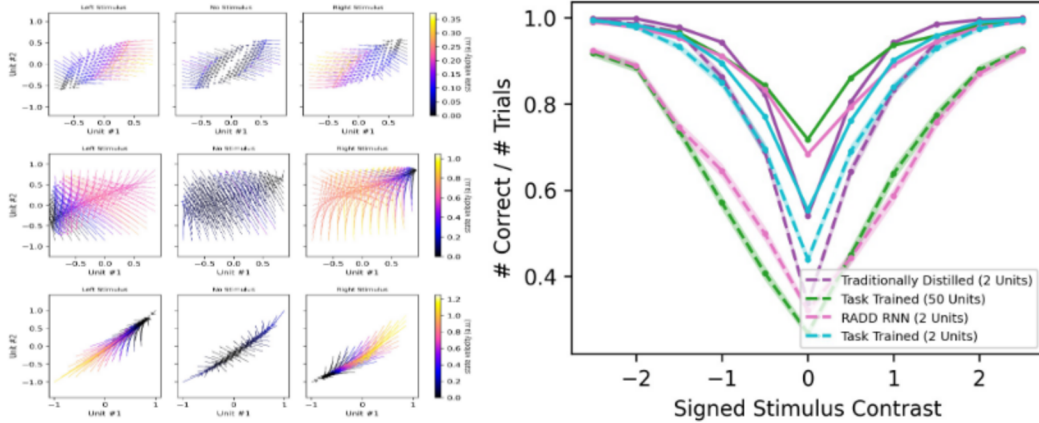

Figure 8: (a) Phase portraits of 2-unit RADD RNN (top) 2-unit conventionally distilled RNN (middle), 2-unit task trained RNN (bottom). Only the 2-unit RADD RNN preserves the phase portrait of the high dimensional, task trained RNN. (b) Comparative performance of 50-unit task-trained RNN, 2-unit task-trained, 2-unit conventionally-distilled, and 2-unit RADD RNNs.

(with rows in $P$ set to the block and trial side readout vectors; similar results are obtained with $P$ set to the first two principal components) closely emulate the original RNN dynamics (Fig. **??**b). In contrast, a 2-unit conventionally distilled RNN and 2-unit RNN trained directly on the task both fail to learn the phase portrait of the original high-dimensional RNN (Fig. 8a), showing that RADD enables the construction of minimal RNN models when other methods do not.

RADD is also much faster to run than conventional distillation since RADD has a closed form solution that does not require gradient descent. RADD took less than a minute of clock time to complete, whereas conventional distillation required several hours ($\sim$5) to complete 10,000 gradient steps (SI 7.4), and sometimes failed to converge even after 10,000 gradient steps (SI 7.5). Depending on the magnitude of the RADD RNN's readout vector (a free parameter), the 2-unit RADD RNN can slightly outperform the full RNN (distilled 86.87%, full 85.50%) on the task (Fig. 8b).

The RADD RNN (Fig 9a) is highly interpretable: activation of its two units correspond to stimulus and block side beliefs, respectively, with recurrent interactions that support integration:

$$\hat{z}_{n,t} = \begin{bmatrix} \text{Stim Belief}_{n,t} \\ \text{Block Belief}_{n,t} \end{bmatrix} = \tanh\left( \begin{bmatrix} 0.54 & 0.31 \\ 0.19 & 0.84 \end{bmatrix} \hat{z}_{n,t-1} + \begin{bmatrix} -0.20 & 0.20 & 0.005 \\ -0.04 & 0.04 & 0.02 \end{bmatrix} \begin{bmatrix} o^L_{n,t} \\ o^R_{n,t} \\ r_{n,t} \end{bmatrix} + \begin{bmatrix} 0.00 \\ 0.00 \end{bmatrix} \right)$$

The input weights show that observations drive the stimulus and block side beliefs in a common direction, but that the movement is 5 times greater along the stimulus direction than the block direction. The self-excitation of each unit extends its stimulus integration time (for an effective time-constant of $\tau/(1 - w_{self})$; the two effective time-constants reflect the observed differences in integration timescale for stimulus and block beliefs. The excitation between units shows that the stimulus belief and block belief reinforce one another. Finally, the feedback input receives negligible weighting, consistent with our earlier observation that the network disregards feedback.

We returned to the task-trained RNN to see whether we could extract the same two-variable circuit seen in the RADD network. We ordered hidden units using hierarchical clustering with Pearson correlation of RNN activity, as the similarity metric revealed two clear subpopulations (Fig. 9b). Units within each subpopulation are strongly correlated, and strongly anticorrelated with units in the other subpopulation. Applying the same ordering to the recurrent weight matrix revealed self-excitation within and mutual inhibition between subpopulations (Fig. 9c). Although superficially appearing to conflict with the RADD circuit (Fig 9a), they are not in conflict; mutual inhibition between block and stimulus populations(Fig 9d) is a form of positive feedback, just as mutual excitation is. They become mathematically equivalent if the signs of one of the two populations is reversed.

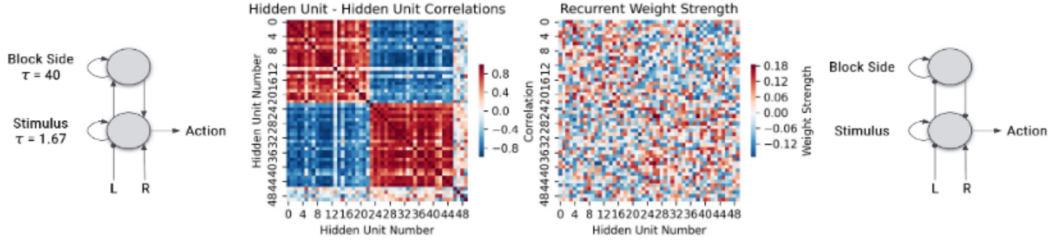

Figure 9: (a) Circuit of 2-unit, RADD distillated RNN that produces both line attractor and bistable attractor dynamics, with different integration timescales. (b) Ordering RNN units based on Pearson correlation of activity reveals two anticorrelated subpopulations (pools). (c) Applying the same activity-correlation ordering reveals self-excitatory, mutually-inhibitory connections between pools. (d) Circuit of task-trained RNN, with each unit now depicting a pool. This circuit is mathematically equivalent to the 2-unit RADD circuit in (a), with mutual excitation between the stacked integrators replaced by mutual inhbition, both forms of positive feedback.

## 4  Discussion

There is a significant body of work on dynamics and inference in RNNs e.g. [18, 37, 26, 38, 45, 35, 25, 39, 29, 30, 22, 6, 11]. Our paper is similar to [34], who also seek to characterize how RNNs perform Bayesian inference on a different task. Our model interpretation technique, RADD, builds on previous distillation work. We differ by proposing supervised learning should occur on the internal activity of the teacher RNN rather than its readouts [7, 4, 14] or a beam-decoder sequence [19]. Our approach is similar to [21] and [12], who both use one recurrent network to train another. However, their focus is on taming chaotic dynamics rather than model compression, and our networks do not receive the desired recurrent activity as an input.

In conclusion, RNNs attain near-optimal performance on a hierarchical inference task, as measured against Bayesian observers and actors that have full knowledge of the task. We have characterized the RNN's representations, dynamics, and mechanisms underlying inference. In future work, we will leverage these models, along with the work of others, to better understand mouse behavior and neural representations. We expect exploring RADD in reinforcement learning [31, 9] will be fruitful.

## 5  Acknowledgements

We would like to thank the IBL Theory Working Group, MIT's Nonlinear Systems Lab (Nick Boffi, Carlos Barajas), and the Fiete Lab for helpful discussions and comments on the manuscript.

## 6  Broader Impact

We appreciate NeurIPS asking researchers to evaluate the ethical dimensions of their work. We believe this work, focused on understanding mechanisms of how neural networks solve basic inference problems, does not have detrimental social ramifications. It is possible that RADD, as a technique for more interpretable ANNs, could be used to better understand biases learned in RNNs.

## 7  Funding and Competing Interests

MK is funded by the MIT McGovern Institute, LM by the Schwarz Foundation and IRF by the Simons Foundation through the Simons Collaboration on the Global Brain, by HHMI through the Faculty Scholars program, and by the Office of Naval Research through N00014-19-1-2584. The authors have no competing interests to declare.

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
