[Supplementary Material]

# Supplementary Information

## 7.1 Analysis and Figures Details

Analyses and figures were generated using network behavior and activity over 1000 blocks $\approx 50,000$ trials $\approx 175,000$ RNN steps. We generated the network behavior and activity from one long continuous session instead of multiple parallel sessions as a precaution against possible initial transients, although we never saw anything to suggest such transients existed or could affect our results.

## 7.2 Training Implementation Details

At the start of each gradient step, the RNN hidden state was reset to the 0 vector. The RNN then proceeded to complete a single session (equivalently, batch size 1) over 4 blocks with an average of 50 trials/block, where the block sides, stimulus sides and observations are randomly sampled according to the task's generative model. The batch size, number of blocks, and trials per block were chosen once and worked well enough, so we did not perform any hyperparameter tuning on them. Gradients were backpropagated through all RNN steps in the session (4 blocks per gradient step * $\sim$50 trials per block * $\sim$3.5 RNN steps per trial = $\sim$700 RNN steps per gradient step). We did not use truncated backpropagation through time; given the performance of the trained network, it appears that exploding and vanishing gradients were not a problem. The cross entropy loss was computed at each RNN step, where the target is the current trial's stimulus side (left or right):

$$\mathcal{L} \overset{\text{def}}{=} - \sum_{n=1}^{N} \sum_{t=1}^{t_{action}} \delta_{s_n,-1} \log(a_{n,t}(1)) + \delta_{s_n,+1} \log(a_{n,t}(2))$$

where $n$ is the trial number, $t$ is the time-step within the trial, $t_{action}$ is the time-step at which an action is taken and the trial terminates, $s_n = -1$ for a left stimulus side and 1 for a right stimulus side, $a_{n,t}(1), a_{n,t}(2)$ are the probabilities of a left and right action and $\delta_{ij}$ is the Kroencker delta ($\delta_{ij} = 1$ if $i = j$, 0 otherwise). At the end of each trial, we compute the feedback signal by comparing the action taken with the stimulus side. We designed the feedback signal to emulate both the external and internal rewards (waiting pressure) experienced by the animal performing the task. We found that the feedback signal has no influence on network dynamics after training (Figure 7c). (It will be interesting in future work to explore whether changing the waiting penalty influences the training trajectory itself even if it has no influence after training.)

PyTorch and NumPy random seeds were both set to 1 for reproducibility (this had no discerable effect). We used stochastic gradient descent with initial learning rate 0.001, initial momentum = 0.1 and no weight regularization. RNN parameters were initialized using PyTorch defaults. These hyperparameters were not tuned. All models we trained converged in approximately 5000 gradient steps and their behavior did not change significantly when trained for longer (up to 100,000 gradient steps).

## 7.3 Representation and Dynamics Distillation Implementation Details

At a high level, RADD consists of training a student network to match the projected activity of the high dimensional teacher network. Consider a trajectory $\{h_t, o_t\}_{t=1}^{T}$ from a dynamical system

$$h_t = f(Ah_{t-1} + Bo_t + c),$$

where $h_t \in \mathbb{R}^N$ is an $N$-dimensional state vector, $A \in \mathbb{R}^N \times \mathbb{R}^N$ is an recurrent matrix, $B \in \mathbb{R}^N \times \mathbb{R}^Q$ is an input matrix from some $Q$-dimensional space, $c \in \mathbb{R}^N$, $f$ is a invertible pointwise nonlinearity, and $T \gg N$.

We define a projection $P : \mathbb{R}^N \to \mathbb{R}^M$ to map from the dimension of the teacher network to the dimension of the student network. We define the target trajectory as the projection applied element-wise to the original trajectory i.e. $\{z_t, o_t\}_{t=1}^{T} \overset{\text{def}}{=} \{Ph_t, o_t\}_{t=1}^{T}$. In our paper, $P$ is comprised of the two readout vectors (the stimulus readout vector and the block readout vector). We then learn parameters $A' \in \mathbb{R}^M \times \mathbb{R}^M, B' \in \mathbb{R}^M \times \mathbb{R}^Q, c' \in \mathbb{R}^M$ such that the dynamical system

$$\hat{z}_t = f(A'\hat{z}_{t-1} + B'o_t + c')$$

starting from initial condition $\hat{z}_1 = z_1 \stackrel{\text{def}}{=} Ph_1$ produces a trajectory $\{\hat{z}_t, o_n\}_{t=1}^T$ with $\hat{z}_t \approx z_t$ across $t$:

$$\underset{A',B',c'}{\arg\min} \sum_t ||z_t - f(A'\hat{z}_{t-1} + B'o_t + c')||^2,$$

Since $f$ is invertible, we can instead optimize:

$$\underset{A',B',c'}{\arg\min} ||f^{-1}(z_t) - (A'\hat{z}_{t-1} + B'o_t + c')||^2.$$

This is now a linear least-squares regression problem, so we solve for $A'$, $B'$ and $c'$ using a conventional OLS solver. Another view is that backpropagation through time combined with (projected) teacher forcing [43] is equivalent to linear regression.

## 7.4 Traditional Distillation Implementation Details

Traditional distillation [7, 4, 14] trains a student network on the output probabilities/logits of a teacher network. For comparison with RADD (SI 7.3), we created a traditionally distilled RNN. We first instantiated an RNN exactly matching the RADD RNN (i.e. 2 hidden units), with parameters initialized using PyTorch defaults. We then trained the new RNN to minimize the cross entropy between its output probability distribution and the output probability distribution of the high-dimensional, task-trained RNN. We trained for 10,000 gradient steps (twice as many gradient steps as than the original task-trained RNN took to converge), where each gradient step used a freshly sampled session of the task to make overfitting impossible. We used stochastic gradient descent with initial learning rate 0.001, initial momentum = 0.1 and no weight regularization. RNN parameters were initialized using PyTorch defaults. These hyperparameters were not tuned.

## 7.5 Differently Sized Networks

The RNN described in the paper has 50 hidden units. Here we show that RNNs of different sizes (25, 100, 150, 250) find similar solutions. As with the RNN described in the paper, random seeds for both NumPy and PyTorch were set to 1.

# Figure 10: 25 unit RNN

# Figure 11: 100 unit RNN

# Figure 12: 150 unit RNN

Figure 13: 250 unit RNN