[Reviews · NeurIPS 2020]

Review 1

Summary and Contributions: In this work the authors trained RNNs to solve a task that requires to infer and represent two latent variables that evolve on two separated time-scales. They show that the behavior of RNNs is quantitatively similar to optimal Bayesian baselines. They then reverse-engineer the trained RNN using various techniques to provide a detailed dynamical system description of how the two latent variables are represented in network's hidden state, how they influence network's decision, and how learnt connectivity support this dynamics. Finally they employ a distillation technique to recover some of the insights they gained from the previous reverse-engineering work.

Strengths: - This high quality work provides a dynamical system perspective of Bayes optimal computations in RNN. They do so for a task involving inference of a latent variable over a long time scale, which advances our understanding of context-dependent computation in RNN. - The task is of interest for neuroscience as part of the IBL project.

Weaknesses: - It is not clear to what extent this is relevant to the non-neuroscientific crowd at NeurIPS as the part about the distillation technique is a small portion of the work and not completely novel as stated by the authors. I thus wonder whether this work would be better suited for publication in a neuroscience journal.

Correctness: Yes

Clarity: Yes the paper is well written.

Relation to Prior Work: It would be nice to discuss the relationship with this work, that reverse-engineer RNN performing Bayes optimal integration of stimuli: Bayesian computation through cortical latent dynamics H Sohn, D Narain, N Meirhaeghe, M Jazayeri Neuron 103 (5), 934-947. e5

Reproducibility: No

Additional Feedback: Update: I thank the authors for their update, that will surely improve the quality of the manuscript, and for making their code available. I thus keep my score at 7 as this paper will make for a nice contribution. The only major concern I have regards the appropriateness of the work for the NeurIPS audience. Other than that I found the work interesting and of a very high quality. One thing I found missing in the paper are details about the training procedure. In order to infer the values of the latent variable corresponding to block identity, RNN need multiple trials to do so. In a typical RNN training, for neuroscience tasks, gradients are back-propagated only over one trial, and network's state is drawn randomly, or at 0, at the beginning of each trial. Are gradient back-propagated over multiple trials ? If so, what about the vanishing gradient problem ? How are network states initialized at the beginning of each trial ? Another question related to training relates to the reward input (it is mentioned not to be important after training but I guess it is instrumental for the training part). This reward input depends on the trial by trial value of the readout unit and as such can not be defined a priori (typical training procedures define a set of input-output a priori, or use RL like approaches). How is this problem overcome here ? Another improvement would be discuss/motivate a bit more the distillation approach, e.g. which new insight does it give compare to the previous analysis, or which insight it does not give compare to the previous analysis ?


Review 2

Summary and Contributions: The authors train a RNN on a task inspired by an experimental neuroscience task. The task requires integrating information across two timescales. The authors analyze the RNN in a variety of methods. Performance is compared to ideal Bayesian solvers. The internal representation of the RNN is analyzed in a meaningful fashion. RNN dynamics show a line attractor that explains the observed representation. Finally, a novel pruning method reduces the RNN to a 2-unit network that retains its main properties.

Strengths: 1. A complete analysis trajectory of a neuroscience-relevant task by an RNN. From training, through behavior and novel analysis techniques. 2. Introduction of the reduction method.

Weaknesses: 1. There are no statistics. It seems that the entire paper presents a single instance of a trained RNN. 2. The clarity of the paper – figures and explanations of methods – is lacking.

Correctness: The results appear correct.

Clarity: The rationale and results are clearly presented. Figures have a very small font size, and the colors are sometimes highly similar (and not colorblind friendly). Some of the details are unclear – is the objective to increase reward? Is training done in batches? Cross entropy with regards to which target?

Relation to Prior Work: There is no dedicated section for related work, and the discussion section does not address that either. The introduction mentions related work, but without clearly comparing the present work to them. A few related works that seem relevant: 1. Full-FORCE has internal activity as training targets. This is related to the model compression 2. Tasks involving two interacting variables (Line 187). Mante & Sussillo have context interacting with stimulus. Sussillo&Barak 2013 has a plane attractor. Romani & Tsodyks (Plos comp 2010) study two coupled continuous attractors.

Reproducibility: Yes

Additional Feedback: POST REBUTTAL EDIT: I read all reviews and the author response. Clarity and lack of details were a major issue - and it seems that the authors will amend this. Most of my concerns in this regard were lack of details, and therefore I am satisfied with the proposed edits. Statistics - the authors state that they checked four networks but do not plan to do any statistics. I think statistics would greatly strengthen the paper, and urge the authors to do them for the final version. I increase my score from 7 to 8 ------------------------------------------------------- A few additional comments: 1. Panel labels (A,B,C) are missing from all figures. 2. What is Tmax? 3. Why the choice of the small waiting penalty? How does it affect behavior? 4. L91: fully specified unless there is timeout. 5. 3.1.1. the accuracy of the Bayesian agent is affected by the threshold. What is the threshold? How does it affect accuracy? How was it chosen? 6. Fig 2A: Marking chance level could help. 7. Fig 2C: It’s quite hard to see what’s going on there. 8. Figure 3B: Perhaps it would be better to only show the zero stimulus case. 9. Figure 3C: not described in the text. 10. L117: “additionally…” Not entirely clear what this sentence adds on top of the previous one. 11. L121 “approaches” in what sense? Is there some limit? 12. Figure 4A: The dots are clearly separated with a large margin. Why isn’t performance 100%? 13. Figure 4A: The description is very partial. What are the points? Were they taken from a specific timepoint in the trial? Which? 14. Figure 4B,C: A colorbar could help. Are there less trials for the right block? 15. L146: showing a time point. Which time point? 16. Figure 5B: very light colors. Hard to see. 17. Figure 5C: Why the sudden increase at the last trials? 18. Figure 7B: It’s hard to see the pattern. Perhaps add distributions of values for each quadrant. 19. Figure 7C. Color is not explained. I assume this is the norm of the flow of dynamics, similar to the measure in Sussillo&Barak (2013), but this is not explained well. 20. L217: The “Delta timestep decoherence”. I’m not sure this is a standard term. If this is where it is being defined, the sentence doesn’t make it clear. 21. Figure 8B: Hard to see because curves are on top of each other.


Review 3

Summary and Contributions: The paper extracts the mechanism by which an RNN implements a complex task, involving accumulation of evidence on two different timescales. Behavioral analyses show that the RNN approaches optimal bayesian behavior. Dimensionality reduction analyses point to a dynamical mechanism reminiscent of line attractors, with stimulus posterior and block posterior being encoded along 2 different directions. Finally, the authors apply a novel model compression method to further clarify the mechanism. Overall, the paper both introduces a very promising method for reverse-engineering RNNs.

Strengths: - exhaustive theoretical analysis and interpretation of a trained RNN - focus on a highly relevant neuroscience task, for which large amounts of data will be collected. - beyond this specific task, the method for interpreting RNNs is highly relevant to the NeurIPS community

Weaknesses: - the most interesting part of the paper (analysis of the distilled network) is highly compressed. The corresponding sub-sections are hard to follow. Supplementary information would have been very useful. - it would have been nice to link the final distilled model to Bayesian aspects of the computations. The reader is left wondering how Bayesian integration emerges from the RNN. - it would have been interesting to link the distilled network parameters to the original networks dynamics (timescales of integration and decay), and connectivity (eigenvalues...).

Correctness: yes

Clarity: Overall clear, but some details are missing. Is the beginning of the trial indicated in any way to the network? Presumably there is no reset (otherwise blocks would not be detected), but then does the action on the previous trial bias the action on the next trial beyond what is expected based on block structure? L194: "representation of the block side must be continuous..." - I found the argument difficult to follow Figure text is very small. Labels for panels are missing throughout. Fig 3a: How is concordant/discordant defined for a zero contrast stimulus? Fig 4a: what does every point represent? What are lines and vectors? Fig 4b,c are hard to read Fig 7a: this is the Pearson correlation of which quantity? Fig 7b: the E-I population structure mentioned in the text is not obvious in the figure. Since it is anyway not used for the mechanism, I would remove this panel. Fig8a: not clear what is plotted here Fig8b: "trajectories closely match" -> very hard to see this in the figure Fig 8c: same question as in Fig 3a Several of the references are incomplete (journal, year ... missing).

Relation to Prior Work: Clearly discussed.

Reproducibility: Yes

Additional Feedback:


Review 4

Summary and Contributions: The paper studies the mechanism of an RNN model trained on a block-structured hierarchical inference task for rodents that's centered around inferring two correlated task variables: block side and stimulus side (requiring inference timescales separated by some orders of magnitude). The RNN is compared to two (nearly identical) bayesian baselines to serve as upper performance bounds. Both baselines have two modules that perform posterior inference for the two variables (stimulus side, block side) with full task knowledge, with only one allowed to make an action and move on to subsequent trials. Contributions (and/or findings): 1) RNN model performance comes close to the bayesian upper bound 2) RNN infers the block structure of the trials, and not just the stimulus 3) RNN dynamics collapse to a 2D subspace, governed mostly by the two variables 3) RNN accumulates evidence to infer both variables 4) RNN consists of two anti-correlated populations (encoding left and right) that likely self-sustain own dynamics 5) Authors introduce a distillation method that forces student RNN states to be low-dimensional projection of teacher RNN 6) A 2-unit distilled RNN learns the same variables as the teacher RNN, though a 2-unit RNN that's trained from scratch doesn't learn well

Strengths: - Empirical evaluation is very satisfactory, the claims are quite easy to agree with and are based on the right experiments. - The paper draws similarities between standard RNNs with bayes-optimal actors, offers a thorough slicing of RNN mechanics along multiple dimensions that allows insights into their behavior, and communicates basic tools that could be of use to the broader community in investigating neural networks. Along these axes, it's a worthwhile contribution to the commmunity.

Weaknesses: The distillation objective proposed is lacking some context. It is acknowldged that the proposed scheme is similar to the one employed in tinyBERT; on the other hand, there are other distillation schemes for RNNs that could have been used as baseline, such as sequence-level distillation (https://arxiv.org/pdf/1606.07947.pdf). If a method is posed as a contribution, it's important to visit existing techniques before introducing a new one.

Correctness: Empirical methodology is sound, with the experiments more or less straightforwardly leading to the stated conclusions.

Clarity: The paper is very well written, with some pieces likely missing that could leave a question mark for most. For instance, what was the data to train the RNNs with the CE loss? Is it LM-style teacher forcing (with a predefined sequence shifted between inout and output), or something else? This seems like an important detail left out. Since the codebase stated in the paper is not yet existent, there doesn't seem to be a clear way for the reader to get an answer.

Relation to Prior Work: There isn't much discussion of prior work, or work that moves in similar directions, from different fields (analogies between RNNs and optimal decoders, other works that use artifiical networks to model tasks like these, etc.), and this is a shortcoming.

Reproducibility: No

Additional Feedback: I'd have liked to see the codebase (reproducibility is highly desired), the details on RNN training, relevant work (doesn't need to be competing work, additional context is always very useful), and a distillation baseline if the distillation method itself is considered a contribution. I firmly believe there is great value in disseminating this work, and I'd surely consider adjusting my score upon provision of above information. Additionally, the distillation objective introduced surely has its broader impacts, which should ideally be addressed. ///////////////////////// Update after rebuttal ///////////////////////// I'm satisfied with the rebuttal. I'm raising my score since the authors agreed to provide the requested information and also made their code public. I also advise revising the broader impact section, adding discussion around the introduced distillation technique and its impacts.

[Author Response · NeurIPS 2020]

We thank all four reviewers for their thoughtful and valuable feedback. We agree with all suggested edits and have incorporated the feedback to improve the content and clarity of our paper. Key changes to the paper include:

1. Switched the code repository from private to public.

2. Added a Supplementary Information section, which includes:

   - Implementation details for training and analyzing the RNN. We provided details many reviewers had asked for, including the exact loss function, when the RNN state was reset (only at the beginning of a gradient step), how gradients were backpropagated (over all RNN steps i.e. across multiple trials and multiple blocks), how the feedback signal was computed (at the end of each trial).

   - Distillation implementation details for our Representation and Dynamics Distillation (RADD) technique

   - Results for networks of different sizes (25, 100, 150, 250). Although we didn't perform statistical analyses comparing different networks, the results we discuss in the main body all hold.

3. Added a Related Work section to the main paper, citing additional related work including the references noted by the reviewers, and clarifying the relationship of that body of work to ours.

4. Added a comparison of RADD to traditional distillation, including a comparison of training efficiencies (wall-clock time) for model compression. We also compared the original network's timescales of integration/decay and eigenvalues of the recurrent Jacobian to those of the distilled network.

5. Clarified, added, removed or otherwise edited parts which were specifically mentioned in the detailed review comments.

6. Changed figure color schemes to be accessible to color-deficient readers.

A concern voiced by one reviewer was that, though of good quality, this work might not be a good fit for NeurIPS. However, quoting from NeurIPS's website, "The purpose of the Neural Information Processing Systems annual meeting is to foster the exchange of research on neural information processing systems in their biological, technological, mathematical, and theoretical aspects," which we feel directly concerns this work. Additionally, this paper is similar to recent NeurIPS papers (e.g. [1, 2]), and three of our four reviewers (who we hope are a reasonable proxy for our target audience) seemed to find the work interesting for NeurIPS.

# References

[1] Ingmar Kanitscheider and Ila Fiete. "Training recurrent networks to generate hypotheses about how the brain solves hard navigation problems". In: *Advances in Neural Information Processing Systems* 30 (2017), pp. 4529–4538.

[2] Niru Maheswaranathan, Alex Williams, Matthew Golub, Surya Ganguli, and David Sussillo. "Reverse engineering recurrent networks for sentiment classification reveals line attractor dynamics". In: *Advances in Neural Information Processing Systems* 32 (2019), pp. 15696–15705.


[Meta-Review · NeurIPS 2020]

This is a solid paper that definitely warrants acceptance. The paper is clearly written and makes multiple substantive contributions in terms of training and analyzing RNNs at the boundary of ML and neuroscience. The reviewers identified some issues, primarily relating to clarity and requests for additional details. There is reasonable confidence that updates by the authors will satisfy these requests. There is an emerging field of research involving neural networks being trained to solve tasks that are used in neuroscience experiments to allow comparisons between the representations and dynamics learned by artificial systems and those observed in real neural recordings. This particular paper, while not dealing with real neural data, describes findings that will be of interest for this growing community. In addition, by selecting a task that is of interest for the IBL, it will be relevant for that subset of the neuroscience community. One additional comment in connection to the existing literature. While the task investigated in this work is on the simpler end of the spectrum, thereby permitting comparisons with the tractable Bayes-optimal model, there is other work in a similar neuroscience-adjacent context that emphasizes, for example, how recurrent networks can solve multiple tasks [Yang et al 2019] as well as higher-dimensional control problems [Merel et al 2020]. At present there seems to me to be a bit of a gap between the regime of a single, relatively simple task on the one end and more complex models that are trained to solve multiple, more complex problems. I'm curious if the authors have thoughts on how well their analysis techniques and model compression approach will scale to settings that have higher "task" complexity such as in the examples above. References: "Task representations in neural networks trained to perform many cognitive tasks" Yang et al. 2019 "Deep neuroethology of a virtual rodent" Merel et al. 2020